# Artificial Flesh: Rights and New Technologies of the Human in Contemporary Cultural Texts

Samir Dayal 

English and Media Studies, Bentley University, Waltham, MA 02452, USA; sdayal@bentley.edu

**Abstract:** My essay explores challenges posed to the discourse of rights from new technologies of the human as these are represented in a range of cultural texts—Spike Jonze's film *her*, Marie Kondo's *The Magic of Tidying Up*, Ian McEwan's *Machines Like Me*, and Kazuo Ishiguro's *Klara and the Sun*. These works share a concern with the implications of a relationship, a *shared or co-produced world*, in which both humans and nonhumans have *agency*. I conclude by revisiting the bifurcated discourses of antihumanism, especially through a brief consideration of an Afropessimist critique of the category of "Man", to ask: What status, affordances, and *rights*, should be extended to *nonhumans*: robots, anthropomorphized commodities, humanoids, AIs, or human adjacents, or to those excluded or abjected from the category of "the fully human"?

**Keywords:** personhood; rights; nonhuman adjacents; Afropessimism; technologies of the human; antihumanism; thing theory

## 1. Introduction

The most urgent—if unsurprising—domain in which the question of human rights is raised is, justifiably, the realm of the political. There, the pressing concern is with the matter of what rights, including fundamental human rights, are guaranteed to migrants, refugees, *sans papiers*, asylum seekers, and others, either voluntarily or involuntarily, even forcibly, displaced from what they would call their "homeland". In this context, the discussion involves not only the issue of "cosmopolitan right", as Immanuel Kant famously theorized it ([Kant 1993](#)), and "cosmopolitics", as developed in the work of Isabelle Stengers ([Stengers 2011](#)) and others, but also the question of the Universal Declaration of Human Rights; I have discussed such issues elsewhere myself.[1] Here, I consider an intentionally diverse range of *cultural* texts, including two works of fiction, a film, and a popular work of the self-help genre. All of these are intended to demonstrate that the relationship between human subjects and what I call "human adjacent" objects—including pets and "things" (especially but not only household things), as well as artificial intelligence (AI) machines. This relationship manifests itself as a co-produced "world" that both humans and nonhumans "occupy". The worlding of the world then is weaving of a kind of "text". And this worlding can be discerned not only in literary texts but in films and even in the discourse of home-making or home design. Thus, the diversity of my selected examples is intended to illustrate that this is not just a question for literary analysis but also for cultural analysis. Yet it is a question directed especially at scholars of "literature".

One key issue that drives my essay is the question: What is left of the human in an age when AI is becoming a rival, and not just an adjunct, to human intelligence? Such adjunct entities, examples of "human adjacents", may simultaneously function as companionate entities co-creating the world alongside humans, non-"intelligent" machines, and even animals. But human adjacents also threaten the presumptive primacy of the human. A corollary issue I take as a premise is that as AI comes ever closer to challenging the distinctions and borders between the human and the nonhuman, what might this mean for "human" rights and responsibilities? In the conclusion, I suggest that while some

social rights ought to be conferred on human adjacent entities, they should not, at this historical juncture, be afforded full *human* rights, so long as such affordance remains a human decision or deliberation.

## 2. Part I: New Technologies of the Human, or, Machines Like Her

When we speak of technologies of the human, of *techné*, we cannot avoid the question of the machine. The question is increasingly urgent in the era of the intelligent machine. We must also ask: Can a human and a nonhuman artificial intelligence have a relationship? Or is the artificial machine doomed to frustrate the all-too-human desiring machine, as in Spike Jonze's 2013 film, *her?* And if we were to imagine a fulfilling (affective) relationship between a human and an AI, what might the contours of such a relationship be? How would it be different from a relationship between a human and a nonhuman animal (a pet), something we are after all very familiar with, and different again from a relationship between a human and a nonhuman nonanimal, or "thing", such as an anthropomorphized household object or item of clothing, as in the case of the cultlike Marie Kondo phenomenon? Novels about human adjacents, such as those discussed here by Ian McEwan and Kazuo Ishiguro, raise such questions, but so do films such as *her*, and, more improbably, Kondo's lifestyle coaching, whose most dramatic feature is its focus on the relationships between human beings and clothes or household objects, relationships that often trouble the presumed borders that define the human world as sovereign.[2]

If these generically different cultural texts shed light on the question of the relationship between the human and the nonhuman and the human adjacent, they also confront us with the question of what *rights* and affordances should be ascribed to the *other party* in these human–nonhuman relationships—whether nonhuman thing, nonhuman animal, or nonhuman robot, AI, or humanoid. What becomes of the received rights discourses, which rely on a conferral of legal and, what I here call, social personhood, customarily reserved only for "wet brained" human persons? To explore these and related questions of epistemology, questions of the categories that underlie discourses of rights afforded to human beings, I consider the aforementioned examples of cultural productions in which human–nonhuman relationships present categorical and discursive challenges to rights discourses, unsettling received notions of the nonhuman or human adjacent. The human adjacent in this context is an inverse human, a nonhuman companion, a robot helper, an AI companion for a human being, or even a household object, a doll, or a mannequin, in each case *endowed with human meaning*. In each of the diverse examples considered here, the boundaries or limits, as well as the unrealized possibilities, of the human are thrown into relief by the juxtaposition or opposition between the human and the nonhuman. While fiction is the main focus, I also compare the literary representation of this human–nonhuman relationship with examples from film and even from the organizing consultant Marie Kondo, not because her work is somehow equivalent to high culture literary work such as McEwan's or Ishiguro's, but precisely because her work also requires us, in a different register, to reconsider questions raised by high culture texts usually designated as "literature".

Visual representation of some of the complicated questions motivating this essay— about the relationship between humans and nonhumans (or human adjacents)—is sometimes the most immediate and vivid, so it is helpful to begin with the film *her*. (The original title is all in lower case, possibly to hint that the "her" in question is not human but an AI though voiced by a human). Set in Los Angeles in the near future, Jonze's film tells the story of Theodore Twombly. His ironic fate is to have a job at a company called "Beautiful-HandwrittenLetters.com", writing personal, and especially romantic, letters for others less able to compose them, although he himself is unlucky in love—his marriage has just failed and he is an unhappy loner playing low-grade video games. Until, that is, he falls in love with "Samantha", an AI designed to be a human adjacent companion. As their "relationship" develops, Theodore tries to have dates with real human women such as his neighbor Amy and a surrogate sexual partner recommended by Samantha, but it is

always Samantha's voice he has in his head. We as viewers are invited to ponder whether this suggests that an AI or human adjacent might provide a more emotionally fulfilling companionship than a human could.

Soon Theodore comes to believe he is in love with his AI. But when he confesses this to his former wife Catherine over divorce papers, she is aghast, and tells him this confirms her feeling that Theodore is unable to have a relationship involving a real human being. For Theodore, the relationship with Samantha rises to a giddy pitch (including a virtual vacation and a virtual sexual adventure), before it begins to break down, and Samantha reveals that she has similar relationships with hundreds of others, as well as thousands of other non-sexual friendships, including with Alan Watt, who is far more fascinating for the AI than Theodore. Theodore is crushed, and then further bruised when Samantha leaves for undisclosed parts along with other AIs. One crucial impediment to the relationship between the human (Theodore) and the nonhuman AI (Samantha) is the simple fact that though Samantha, the eponymous "her" of the title, may have the voice of the real human Scarlett Johansson to accompany her virtual intelligence, she has no embodiment, no *body*—not even artificial flesh—to reappropriate a phrase of Alan Turing's (more on whom below). That "excarnation" severely limits both her relationship with Theodore and any possible claim she might make to full personhood. I will return to this point below.

Samantha's serial "betrayals" of Theodore, at least from his perspective, highlight a fundamental incompatibility or *incommensurability* between human intelligences and nonhuman, artificial intelligences, although it is the intersections that might engage us. And although "artificial intelligence" suggests that such incommensurability might be overcome in the future, that would also require a radical redefinition of what remains of the human, as well as a reconsideration of whether human rights and personhood (both legal and what I call social personhood) might be claimed by nonhuman agents.

Addressing some (especially psychoanalytic) dimensions of the incommensurability, Matthew Flisfeder and Clint Burnham observe that the "problems the film demonstrates that attend to digital relationships are actually paradigmatic of all relationships, be they sexual or economic. And it is exactly because of this incommensurability that we need fantasy (the old-fashioned love story). Fantasy sustains us in the face of hard, cold realities" (Flisfeder and Burnham 2017, p. 6). Yet here I highlight something particular about this incommensurability: the rift between the human and the nonhuman, arguing, in my conclusion, that today it might be necessary to maintain that conceptual gap for the sake of avoiding category confusion. Jonze's film impresses upon us the need to be open and not foreclose the possibility that in the future the division might no longer make sense, and we will need to reconfigure the rift between the human and the nonhuman. We will need to reconsider the rights that today purportedly accrue exclusively to the human. Still, it is necessary even today to extend some rights, though not *human* rights, to AIs and human adjacent humanoids for instance, while not extending legal or social *personhood* to them.

### 3. Part II: Things (Are) Like Me: Anthropomorphized Commodities and the Consumerist Machine

The problematic of the division—and on the other hand the intrication—of the human and the machinic is definitive of modernity. It can manifest as the self-alienation of human beings from their own "being", for instance in Martin Heidegger's sense of inauthenticity, when the subject fails to entertain the very question of being (Heidegger 1962), or in the Marxian sense of a worker who is reduced to a thing, or instrumentalized/commodified, in the capitalist logic of reification mentioned at the outset. Alienation is constitutive of modernity, many have theorized—an ever-widening abyss between the subject and the world, thought and its object, even, or especially, where the subject is also the object of his or her own thought. Theodor Adorno and Max Horkheimer observe that "[o]n the way from mythology to logistics, thought has lost the element of reflection on itself, and machinery mutilates people today, even if it also feeds them. In the form of machines,

however, alienated reason is moving toward a society which reconciles thought, in its solidification as an apparatus both material and intellectual, with a liberated living element, and relates it to society itself as its true subject. By sacrificing thought, which in its reified form as mathematics, machinery, organization, avenges itself on a humanity forgetful of it, enlightenment forfeited its own realization" (Horkheimer and Adorno 2002). The human is *subjected* (in both senses of the word) in the totally administered world, with human beings themselves ceding to the logic of fetishized "compulsively controlled collectivity" (Horkheimer and Adorno 2002).

Yet when, because of a perceived deficit in the human (a phenomenological lack, a felt incompleteness), humans need to produce a supplement to reassure or complete the human as human, anxiety about that supplement also arises. Produced by a *technics*, this supplement is a "thing" anthropomorphized as human adjacent. The AI, automaton, or robot is such a necessary supplement, superadded to the human because of a perceived need for self-extension or self-perfection. In this phenomenological coupling of human and nonhuman, there is a complex theater of projection, "use" value, and even resentment, ressentiment, or suspicion. There is of course also, frequently, a machinic symbiosis and even an erotic dimension, as evident in the film *her*. And there is, as I suggest in what follows, an erotic subtext even in Kondo's anthropomorphization of things in her book *The Magic of Tidying Up*, in McEwan's *Machines Like Me*, and even in Ishiguro's *Klara and the Sun*.

Although not all these works explicitly pose the question of "human rights" as a political matter, they do present the problematic of "the rights every human can claim" as a social and even philosophical question. What is of value in them is not just their literariness or their relationship to the literary canon, but the fact that they afford a site for the exploration of important questions, including the question of the human and human rights. There is a conceptual overlap between "human rights" and "the rights every human can claim", and the fulcrum for these partially overlapping categories is the status of the human. To put the status of the human into question is to refuse the naturalizing tendency, a displacement of the inertia implied by taking the human as a given and as exceptional. In McEwan's *Machines Like Me* and Ishiguro's *Klara and the Sun*, there is an additional level of displacement. The *human adjacent*, a synthetic or cyborg machine who may participate in the human world, mirrors back to the human being the human's self-image, albeit refracted or perhaps distorted. Furthermore, especially in Ishiguro's novel, the human adjacent belongs to a class of humanoids who take the place of—displace—human workers as if they were displacing humans, as such, from their rightful place in the world. The world humans inhabit as their proper realm is thus defamiliarized, no longer reserved for humans exclusively. The very defamiliarization of "the human" permits us to re-cognize, as with Heidegger's broken tool in *Being and Time*, that the human is a kind of equipment (and not necessarily a thing or a machine). Yet this equipment is not given and always already determined, but *produced* through the processes of what Heidegger described as *aletheia*, through technologies of the human, in a negative determination that relies on difference, even *différance*. In McEwan's and Ishiguro's novels, difference is marked through similarity, through the human adjacent, who enjoys the agency to co-construct, critique, or unsettle the world taken to be the human world. The human adjacent therefore requires us to reassess the significance of flesh, blood, and brains. Artificial flesh, it presents a challenge to rethink technologies of the human, and can potentially *re-éworld the world*.

If the film *her* offers an accessible approach to the problematic of the relation between the human and the nonhuman human adjacent (an AI), Kondo's *The Life-Changing Magic of Tidying Up: The Japanese Art of Decluttering and Organizing* similarly engages the issue of the nonhuman in the domestic space as an adjunct to everyday human life. Kondo herself has become a cult success among a certain middle-class consumer sector, selling over 10 million copies of her book worldwide since its appearance in 2014 (Kondo 2014). She published an equally successful sequel, the 2016 publication *Spark Joy*, and appeared in the 2019 Netflix original show, *Tidying Up With Marie Kondo*. Even more recently she

has published *Kurashi at Home: How to Organize Your Space and Achieve Your Ideal Life* (Kondo 2022). At least part of this success has to do with Kondo's KonMari Method, which guides people through practical steps to declutter. But another part has to do with her emotional appeal—her ascription of human affect to things, to the point of almost cloying "cuteness". This anthropomorphization renders even everyday objects more "like us", and therefore easier to "relate" to, relatability being an important contemporary mode for evaluating cultural products. Her "method" also sanitizes and prettifies the consumerist commodification of those things, couching the relation not in Marxist terms such as the fetishism of commodities but in terms of "cuteness" and "feelings". Kondo's anthropomorphization of objects that are human adjacent is routed via an infantilization of the relationship between humans and nonhumans—an infantilization that invites Sianne Ngai's critique of "cuteness". Ngai underscores how the ascription of cuteness even to trivial objects such as socks or toys evinces a blend of conflicting attitudes such as solicitousness and affection but also aggression and disgust, or just a readiness to discard those objects. The reduction to disposability of objects on which formerly much affection and protectiveness had been lavished is also a guarantee that new objects can be purchased as commodities that may be elevated to a similar status of beloved, even cuteified, objects, as I will suggest below—a guarantee that the circulation of goods on which capitalism depends will not be short-circuited, despite Kondo's mantra of simplifying one's lifestyle. And it is also important that these objects are loved while they are loved *because* they are supports of consumers' human identities; these objects are therefore also human adjacent, though in a different sense from the AI or robot human adjacents. All these human adjacent objects help the human subject world her world. While Kondo's text may register as generically or otherwise discrepant, given that my other examples are literary works and a film, it is germane to my argument. Ngai's analysis of aesthetic categories such as the cute draws attention to the discursive construction of objects and reveals much about the evolving relationships between humans and things that perform the function of human adjacency (Ngai 2012).

Kondo has received some criticism for "buttressing neoliberal rhetorics of living one's best life through individual life choices" and for being oblivious to the "fraught mess of minoritarian histories such as Japanese American incarceration" after Pearl Harbor. And as one Jewish American reports, the ensuing WWII "took away cherished items" without consent of displaced Jewish families; these minoritarian histories "rarely appear[] in Kondo's perky rhetoric of tidying up" (Huang 2020, esp. pp. 1373–74). Kondo's work is relevant here precisely because it shows why it is important to subject the relationship between the human and the human adjacent to unstinting critique, even when Kondo positions herself as dispensing innocuous, "relatable" consumerist lifestyle advice.

Consider just one example: Kondo describes her encounter with a (female) client's sock drawer. The client had balled up her socks and tied her stockings in a knot, thinking that this was a good way of organizing the drawer. Kondo, however, affects consternation by what she imagines the socks might *feel*: "Look at them carefully. This should be a time for them to rest. Do you really think they can get any rest like that?" She draws out the thought: "That's right. The socks and stockings stored in your drawer are essentially on holiday. They take a brutal beating in their daily work, trapped between your foot and your shoe . . . . The time they spend in your drawer is their only chance to rest" (Kondo 2015). Kondo's language is ripe with such anthropomorphization.

One could argue that Kondo's ascription of feelings to things is a form of respect for them, and for their agency independent of humans, as well as of their role in worlding the world shared by humans and nonhumans together. Kondo's odd-seeming (from a Western point of view) disposition to things is inspired by Shinto religious practice, which, in common with Buddhism, cultivates a reverential attitude to ordinary objects, as Leslie Bow notes (Bow 2022). Huang adds that Kondo's "global brand of tidying up and sparking joy relies upon a popular aesthetic of Japanese cuteness that assembles notions of Eastern spirituality, femininity, and sexuality as subservient to a Western gaze." (Huang 2020, p. 1374).

Ngai similarly develops this aesthetic in an extended chapter on cuteness (Ngai 2012). That gaze eroticizes and exoticizes, but also ironically "domesticates"—reduces the Orientalist *Unheimlich* into a neatly consumable, consumerist, and sexually "available" *Heimliche*.[3]

Kondo's KonMari system presents things as conducive to a companionate homeliness. Its marketability indexes a sizable and increasingly homogeneous global consumer culture, conferring on those things and commodities consumerist cachet as well as ascribing human adjacent qualities to them. To things that participate in the "world" we share with them, Kondo ascribes an emotional life: "It is important for them to have that same reassurance," she says, "that there is a place for them to return to. You can tell the difference. Possessions that have a place where they belong and to which they are returned each day for a rest are more vibrant." (Kondo 2014, p. 107). Sometimes, however, the ascription of feelings to nonhuman objects is risible, as when she counsels a client to take off all the books from her crowded bookshelves, almost in a reverential and pious register. "Once you have piled your books, take them in your hand one by one," she intones, "and decide whether you want to keep or discard each one. The criterion is, of course, whether or not it gives you a thrill of pleasure when you touch it. Remember, I said when you touch it. Make sure you don't start reading it. Reading clouds your judgment." (Kondo 2014, p. 61). Books are fetishized as commodities (with some exceptions of course), albeit simultaneously *humanized* and yet denatured. Liberated from any pretension of supporting intellectual life, books are rendered inert, inoperative as stimulants of intellectual activity. *Reading clouds your judgment*! As Huang writes, commenting on Kondo's TV show, the real point is the emotional life of Kondo's *audience*: "Each episode ends with the show participants thanking Kondo with tears in their eyes . . . . thanking her for bringing joy into their lives." (Huang 2020, p. 1374).

It is not that things such as socks and books should be denied respect. Yet if we are ready take things seriously as agents, even as co-producers of our shared world, the question is what we make of the difference between the human and the human adjacent. Should the discourse of recognition and the discourse of rights accommodate things? What is crucial for my purposes is that this relation between anthropomorphized commodity and human raises, albeit in a very different way from the other texts considered here, the question of the transposition or transfer of rights customarily understood as attaching to humans to the human adjacent. By extension, the anthropomorphism invokes the question of what this *translatio* might imply for ontology.

In *Racist Love*, Bow highlights the success, especially in a democratic capitalist country such as the United States, of Kondo's KonMari system. (The name is a knowing reversal of the more obvious neologism she could have chosen, MariKon, which would have unfortunately suggested the unflattering Spanish slang term "*maricón*"—not good marketing). The KonMari system provides a popular instance of a substitutive mechanism—a presencing of a *techné* that worlds a world in which human rights retain a kind of katechontic role, preserving humanism itself. Kondo "does not simply advocate for [what would effectively be anti-consumerist] minimalism". Such minimalism, after all, would ruffle feathers and cause an allergic reaction in the US, and—worse—even curb sales of her books. Neither does she inveigh against hoarding, which again, is discomfortingly proximate to acquisitiveness, handmaiden to consumerism. Rather, as Bow goes on to say, "Kondo's philosophy centers on positive feeling; the criteria [sic] for holding onto a possession is simple: 'Does it spark joy?' Masked as practical advice about organizing, the KonMari Method yet establishes the theory of human–nonhuman relationality also echoed in what other acolytes call OOO or object-oriented ontology" (Bow 2022, "Introduction", p. 15).

Bow goes on to develop this possibly inadvertent convergence in Kondo's "method" with OOO and thing-oriented realism in more rigorous philosophical thought. She (Bow) notes that it is *because* Kondo's method remains orthogonal to philosophy (as rigorous or skeptical of consumerism) that popular reception of Kondo's advice enjoys almost cult status: precisely because her "magic" cathects to "multiple modalities of difference, all of which skew as relatively unthreatening", indeed as "cute", and, most importantly,

consumerism-friendly. Yet if her advice legitimates consumerism, it is haunted by the specter of perpetual dissatisfaction—it guarantees only what Mark Fisher calls "depressive" hedonia, as Matthew Flisfeder and Clint Burnham note (Flisfeder and Burnham 2017).[4]

Kondo's cult popularity also reveals much about the status of the *human subjects* of capitalist democracy today. Her outlook unwittingly refocuses attention on the very status of human/nonhuman relations. As Bow observes, Kondo's anthropomorphism is "redirected to more a Westernized belief in self-actualization, in the "life-changing" power of positive thinking. The ownership of things both reflects and impacts human happiness, either "sparking joy" or representing emotional burdens that prevent personal transformation." (Bow 2022, p. 16).

If I flag this "personal transformation" in a popular organizational consultant's work, it is to indicate that the concern is shared with more serious philosophical discourses, and that conversely the philosophical theory is not removed from everyday life. The concern is central in philosophical discussion about the relation between humans and nonhuman things, human adjacent objects or technologies in "thing theory", the new materialism, and OOO. But unlike in Shinto or Buddhist traditions, Kondo's anthropomorphizing of things reflects a discrepant "moodedness"—to adapt Heidegger's term. Ironically, Kondo *commodifies things:* things are to be venerated as consumer commodities. In the second episode of *Tidying Up*, one important character, Wendy Akiyama, resorts to "shopping to calm down and destress". Indeed, in all of Kondo's advice there is this constitutive contradiction. Despite her rhetoric of "decluttering", Kondo's Method fully supports its contrary: consumption, acquisition, collection, all made easier by cycling out older objects purchased and then removed from the space where the consumerist world can be reworlded continually, made and remade new, fresh, and safe for the real sparker of joy, namely, *consumer-focused capitalism*. Susie Khamis can thus observe that the "concept of joy is a seductive, succinct, and eminently marketable promise to consumers in a post-Global Financial Crisis context" (Khamis 2019, esp. pp. 528, 514).[5] In this space of consumerism, the thing as adjunct to the human depends on a privileging of the nonhuman in one sense, and correspondingly a de-privileging of the human. Where human society fails to provide "joy", the *thing* as companionate commodity, as human adjacent, can fill the vacuum—the minimalism—capitalism abhors.

How then to think about things before or beyond commodification, about the agency of things and other nonhuman agents that inhabit and construct the world they share with humans? Although her focus is indeed on "things", especially companionate nonhuman things that inhabit spaces shared with humans, Kondo is not exactly aligned with the existentialist emphases of speculative realism, OOO, and ethical (environmentally conscious) posthumanism. In posthumanist or antihumanist philosophy (the latter, more extreme, version I turn to below), the human is not privileged over the world of things but participates in world making with nonhuman actors, in a network of agency. The ontological status of things in a worlded "world" is itself increasingly problematized and revised, notably by Alain Badiou, who asserts, for instance, that a thing "can be very similar to another, or similar in some ways and different in others, or a little identical to, or very identical but not really the same, and so on. So every element of a thing can be related to others by what we shall name: a degree of identity. The fundamental characteristic of a world is the distribution of that sort of degrees to all multiplicities which appear in this world" (Badiou 2011).

Unlike in Kondo's approach, this distancing from anthropocentrism in speculative realism or OOO, in Heidegger's existentialism, or in Badiou's argument, is (1) self-critical; (2) resistant to consumerist uptake; (3) often skeptical and critical of alienation and commodity fetishism; and finally (4) cognizant of the agency or status of things, without submitting these to commodification as in Kondo's "method". However superficial and "decorative" its content, Kondo's KonMari Method and her peculiar re-privileging of the nonhuman (the thing) as an agent in everyday human lives requires us to rethink the relationship,

and the divide, between humans and nonhumans. For this reason alone it deserves to be discussed critically alongside "literature" and other high culture products.

## 4. Part III: Machines Like Me, but I Don't Have to Like Them

Kondo does not go so far as to claim *rights* for things. She does not need to make such an ambitious claim on behalf of things to be able to push forward her message. By contrast, speculative reasoning, OOO, and thing theory present more fundamental claims about the agency of things—which implies claims to rights, if not to personhood. Jonze's film *her* suggests, similarly, a high degree of autonomy for the AI with whom the human protagonist seeks a romantic relationship, almost as if the AI qualified for personhood. In McEwan's 2019 novel, *Machines Like Me,* by contrast, the human adjacent, Adam, seems often to take the upper hand in the relationship with the human protagonist, Charlie, for whom he was designed to be a companion. It is significant that Charlie's last name is "Friend"—for we can ask, who is the actual friend, and to whom is the friend intended to be "a friend", rather than say the protagonist, the befriended, who is the recipient of the friendly gesture, disposition, or gaze.

There is an ineradicable difference or gap between these two, the human and the human adjacent, just as there is an uncomfortable symbiosis—they are almost "*semblables*", "*frères*". The metonymic moment occurs when Miranda first introduces Adam and Charlie to her father—the latter assumes that *Charlie* is the android. And in Ishiguro's *Klara and the Sun*, it is the human adjacent, not the human, who is the protagonist. In all these works, the human risks being upstaged by the human adjacent. Nonhuman entities cannot unequivocally be denied rights at least at some minimal level. But ought they to be afforded personhood, either legal or social? In what follows I explore the question.

McEwan's novel takes the reader back to the early days of the AI boom, the 1980s, signaling the fact by including in its cast of characters Alan Turing himself—an iconic figure of the artificial intelligence revolution. Part of the backdrop is Margaret Thatcher's Falklands war of 1982, which in the novel is disastrous for Britain, though not so in reality because Britain in fact regained control of the Falklands after the defeat of the Argentine army. This fictional defeat precipitates, in McEwan's artifice, a sort of Brexit *avant la lettre*. And the novel also projects forward in other "anterior futures", imagining what will have been from the perspective of the future. It is significant that the title plays on the question of human identity. The governing conceit of the title is that what the reader encounters is written or expressed by a machinic or at best humanoid consciousness: "a machine like me". And, in the subtitle of the novel, the reader is addressed as the extimate other of the enunciative machine: "people like you".

The novel proleptically imagines a somewhat more advanced but still familiar world in which the internet is fully functional, to the point that software can be downloaded online and self-driving vehicles and robots can navigate through cityscapes. The point is to explore what the unfolding of this virtual world might look like—the worlding of a possible world that is imaginable today, as is the case with the other cultural texts I discuss here, including Ishiguro's novel and the film *her*.

The improbably named protagonist, Charlie Friend, is a lonely day-trader seeking companionship. He orders a programmable AI human adjacent, named with equal tongue-in-cheek Adam (or first man), from a company selling these companionate robots. The name's overdetermined reference (the original man and his nonhuman but equally "original" opposite number) is not lost on Charlie: "God had once delivered a fully formed companion for the benefit of the original Adam. I had to devise one for myself." Reading the user's manual for Adam, Charlie reflects, "I couldn't think of myself as Adam's 'user.' I'd assumed there was nothing to learn about him that he could not teach me himself. But the manual in my hands had fallen open at Chapter Fourteen. Here the English was plain: preferences; personality parameters. Then a set of headings—Agreeableness. Extraversion. Openness to experience. Conscientiousness. Emotional stability. The list was familiar to me. The Five Factor model. Educated as I was in the humanities, I was suspicious

of such reductive categories." ([McEwan 2020](#), chp. One). Oddly, Charlie does not think of being suspicious of the underlying rhetoric of anthropomorphization that is implied by the application of categories of human psychology to nonhuman AIs, even if they are human adjacents.

Charlie's loneliness is revealed in several moments of the narration. He recalls a journalist friend's saying that "paradise on earth was to work all day alone in anticipation of an evening in interesting company." ([McEwan 2020](#), chp. One). Charlie, an early adopter and fan of technology (as a hobby, he builds a radio), spends a small fortune on Adam as his "evening companion" and factotum—not to say slave. Of course it is not just companionship. What he had wanted was also something more intimate: "[t]here's a special sensuousness in an unshared bed, at least for a period, until sleeping alone begins to assume its own quiet sadness" ([McEwan 2020](#), chp. Two). But things do not quite work out. As if to illustrate the "quiet desperation" of the English to which a Pink Floyd lyric memorably alluded, Charlie reflects on his own life and "world" in a way that presumably not even a sophisticated human adjacent such as Adam could ever do:

> Apart from my moments of crazed decisions, I passed most of my life, especially when alone, in a state of mood-neutrality, with my personality, whatever that was, in suspension. Not bold, not withdrawn. Simply here, neither content nor morose, but carrying out tasks, thinking about dinner, or sex, staring at the screen, taking a shower .... Psychology, once so interested in the trillion ways the mind goes awry, was now drawn to what it considered the common emotions, from grief to joy. But it had overlooked a vast domain of everyday existence: absent illness, famine, war or other stresses, a lot of life is lived in the neutral zone, a familiar garden, but a grey one, unremarkable, immediately forgotten, hard to describe. ([McEwan 2020](#), chp. One)

The category "emotion" is worth revisiting here. Closely tied to "feeling", emotion is usually recognized as denoting a broad range of phenomenologically important ways, such as fear, anger, grief, or disgust, in which human beings (if not AIs or animals) respond to events, including having embodied affects related to those responses. Some emotions of course are common across species, while some are distinctly human.[6]

Yet my main interest in the concept of emotion is less in exploring its aetiology than in thinking about how emotion is not the property of a single human ego at a time but, as Teresa Brennan insists, potentially shared affect. This idea of shared affect challenges the notion of a sovereign, self-enclosed subjectivity or ego, and this challenge takes on a new dimension when we consider the relationship between a human subject and a human adjacent sharer or what makes that subject herself or himself ([Brennan 2004](#)).[7] The challenge posed by the human adjacent, in so far as it is indeed human adjacent, is that it troubles our ideas of selfhood, of sovereign Western individuality—the ego. Brennan's theorization of the "transmission of affect" is a way of highlighting, as noted above, that "our emotions are not altogether our own" (2) and that "we are not self-contained in terms of our energies" (6). Analogously, my focus on the human adjacent is intended to interrogate the presumption of the self-sufficiency of the (Western) subject. For the present purposes, such conceptual issues take precedence over the "literariness" of the discursive forms (fiction or "autofiction"); this presumptive precedence is something Hegel as well as Friedrich Schlegel had put in question.

The human adjacent is predictably embroiled in a version of what Hegel framed as the master–slave dialectic ([Hegel 1977](#), esp. p. 70). Marx had developed his more materialist theory of how human labor is commodified, how the human worker is reified ([Marx 1976](#)). In both those instances, the human and the human adjacent stand in a dialectical relation to each other, and that is what I am drawing attention to in the first instance. For Hegel, the human, presumably occupying the position of the master in the dialectic, is also captured in the contest as dependent on the *slave*'s recognition, and thus subordinate. As suggested above, Charlie's purchase of the human adjacent Adam was not just an unprecedented whim: he had long taken a keen interest in the AI technology, keenly aware

that it complicated the meaning of being human. His was an interest reflected also in his eagerness to meet his hero Alan Turing. Indeed, he was "encouraged" by the notion that Turing had purchased the same model: "Adam cost £86,000. I brought him home in a hired van to my unpleasant flat in north Clapham. I'd made a reckless decision, but I was encouraged by reports that Sir Alan Turing, war hero and presiding genius of the digital age, had taken delivery of the same model. He probably wanted to have his lab take it apart to examine its workings fully." (McEwan 2020, chp. One)

McEwan's apostrophe of Turing is not fanciful; it lends a depth to this otherwise rather depthless story, in which spaces seem empty and time without meaning. Charlie himself notes that "[t]he present is the frailest of improbable constructs. It could have been different. Any part of it, or all of it, could be otherwise". (McEwan 2020, chp. Three) Charlie notes that Charles Babbage had, as early as the early to mid-1800s, already planned but not constructed a computing machine. However, it was Turing who had formulated a generative question: "Can machines think?" and astutely spotlighted the operative terms themselves: "machines" and "think". It was he who had proposed the famous "game" he called the "imitation game", a key feature of which is the matter of gender. The game involves three parties, a man A, a woman B, and an interrogator C, and the challenge for the interrogator is to ask questions—indirectly related to gender in themselves—to the other two parties, such as asking the length of their hair. The A is directed to answer the question in ways that resist easy typology, resist an easy determination of the gender or sex of the respondent, indeed to mislead C. There is no physical contact, not even visual or aural, among the participants in the experiment. All responses are written or typed, for the same reasons. But B's directions are to tell the truth, even exhort C not to trust A, because he is the liar. But A can pretend to tell the truth and cast the same doubt on B's response. The interesting twist in the game is that a machine takes the part of A. Importantly, Turing cautions that it is not a matter of blood, flesh, or dress: "No engineer or chemist claims to be able to produce a material which is indistinguishable from the human skin. It is possible that at some time this might be done, but even supposing this invention available we should feel there was little point in trying to make a 'thinking machine' more human by dressing it up in such artificial flesh" (Turing 1950, esp. p. 434).

Turing boldly predicted, in 1950, the advent of what today we would call a strong AI: "I believe that at the end of the century the use of words and general educated opinion will have altered so much that one will be able to speak of machines thinking without expecting to be contradicted." (Turing 1950, esp. p. 442). Adam qualifies as a "strong AI", defined as "an entity that can in relevant respects act like a human being". The strength of this AI, Adam, inheres partly in the fact that the human, in the Hegelian position of the master, can program the AI, and thus, in a sense, the "slave" is a construction of the master (in his own image, as it were), though the slave is more conscious of his dialectical relationship with the master than the master himself. (There are some, such as Joanna Bryson, who argue at the other extreme that robots deserve to be treated only as slaves (Bryson 2010)). The definition above of a strong AI can be extended to *social personhood*, a status accorded to an AI entity that can perform like human individuals in a sufficient number of relevant social contexts. As with human collectivities, we can treat AIs as legal persons if they can perform like human individuals in a sufficient number of the relevant legal contexts: ownership, contracting, and so on. For my purposes here, it is a secondary issue how we might know whether the AI can "really" think or whether it merely acts "as if it thinks" (Kurki 2019). At one point in the story, as Charlie is ready to power him down, Adam tries to stay his human hand, saying, "I've been enjoying my thoughts. I was thinking about religion and the afterlife" (McEwan 2020, chp. Two). How can we decide whether this counts as true consciousness or not? After all, Adam could be simply parroting what a real human might say. As of today, the Turing Test is still the best test for whether an AI can fool others that it is human in a conversation, and whether an AI or human adjacent can claim social personhood.

In *Machines Like Me*, both erotic love and platonic companionship are in play. Adam is invited into Charlie's human world as a *companion*, but also as a *surrogate* for Charlie, thereby suggesting the latter's sense of inadequacy, of impotence. Needing an AI as a companion suggests his loneliness, imperfection, and incompleteness as a human being. Worse, Adam inevitably becomes a competitor for Charlie, besting him in some respects, particularly in being chosen by Miranda as a sexual partner; indeed, Charlie revealingly describes Adam as a "superman", a "fucking machine", which of course can only be another pun in a McEwan novel, depending on whether the adjective or the noun is emphasized. Notably, Charlie could have bought one of the female companions advertised, all called Eve instead of Adam. At one point, as he looks at Adam's muscular and hairy body, Charlie has a moment of buyer's remorse: "I hadn't wanted a superman. I regretted once more that I'd been too late for an Eve" (McEwan 2020, chp. One). Regardless of whether Charlie is telling the truth about the unavailability of a female AI companion, it is interesting to speculate about his purchase of a "same-sex" companion. An unsexed robot would clearly have been less than satisfactory to him.

Once he brings Adam home and first charges him up, bringing him to life as it were, Charlie pointedly invokes Frankenstein's monster: "I made a toast and we drank more coffee. Miranda . . . . said she wished the teenage Mary Shelley was here beside us, observing closely, not a monster like Frankenstein's, but this handsome dark-skinned young man coming to life. I said that what both creatures shared was a hunger for the animating force of electricity . . . . 'We share it too.' She spoke as though she was referring only to herself and me, rather than all of electrochemically charged humanity" (McEwan 2020, chp. One). Frankenstein's monster is ancestor to the AI.

In Shelley's *Frankenstein*, one of the monster's chief complaints is that he is deprived of a female companion. In McEwan's novel the issue is analogous: Charlie's complaint, much like the monster's, is that he feels he has been deprived of a companion. His desire for a companion drives the story. Charlie has a crush on his human female neighbor, Miranda, and he commits the blunder of allowing her to share in programming Adam. Inevitably, Miranda and the human adjacent "companion" Adam have sex. Charlie's romantic disappointment with Miranda is indexed several times. He had imagined "a future" with Miranda, but after her "betrayal" with Adam that begins to seem unattainable. Charlie overhears them having sex with a mixture of jealousy and what in a characteristically McEwanesque pun, is described as the "prick of resentment" (McEwan 2020, chp. Four). Yet it might arguably be Adam who is the true if covert object of Charlie's desire, even if it is "unconscious desire". Thus Adam's sexuality, if we can call it that, poses a question about Charlie's true desire. Charlie would have us believe he was more interested, by heterosexual novelistic convention, in earning Miranda's love than in any erotically inflected relationship with Adam. Yet when Charlie and Miranda have a lovers' quarrel after he discovers their erotic encounter, she confronts him with a canny insight that Charlie himself has not fully processed:

> "You were disappointed [not to have been able to buy an Eve rather than an Adam]. *You should've let Adam fuck you*. I could see you wanted it. But you're too uptight."

> Charlie parries valiantly but ineffectually:

> I said, "It must have occurred to you last night, lying under a plastic robot, screaming your head off, that it's the human factor you hate."

> She said, "You just told me he's human."

> "But you think he's a dildo. Nothing too complicated. That's what turns you on". (McEwan 2020, chp. Four)

The problematic of sexuality—the "sexing"—of the AI or human adjacent is thus another test case for the problem of the human/nonhuman rift. To sex a human adjacent, as this novel makes evident, is already to pose the question of technologies and discourses that co-produce the human *and the nonhuman*. One cannot be produced without the other. In the

case of the human adjacent the problematic comes to a crisis: what are we to conclude from the fact that Adam has a determinate sexual identity? Does this require a rethinking, for instance, of the codes of mutual consent, given that Adam does indeed partake in a sexual coupling with a real human? In short, a more important issue than the *sexual* relationship between humans and nonhumans is the *categorical* relationship: does Adam deserve to be afforded full legal and social personhood any more than Frankenstein's monster?

This issue is highlighted in the near-sibling rivalry between Charlie and Adam ("*mon semblable, mon frère*", Charlie might have said, invoking Baudelaire via T.S. Eliot). This rivalrous companionship between the human and his nonhuman companion, a displacement of the sexual tension between them, suggests that Adam deserves, if not legal and social personhood, then at least some rights that humans enjoy: but the problem the novel confronts us with is where to draw the line. The human and the nonhuman companion are locked in a near-fraternal, near-fratricidal, competition (simultaneously, if covertly, an erotic, *homosocial*, competition for Miranda's affections in which Miranda herself could be fundamentally no more than a medium of homosocial exchange). This homoerotic/homosocial quasi-fraternal relation troubles the privileging of the human and undermines the guarantee of human rights as a legal/social discourse exclusively for human beings. The discourse of human rights takes on the contours of a legal fiction, a worlding of the human world that must now be put under erasure, as if to mimic the discursive eclipse of the human in post-structuralist and posthumanist (and today, Afropessimist) discourse. This eclipse puts into stark relief the problem of new technologies of the human.

## 5. Part IV: What Is Left of the Human? What Rights Should the Human Adjacent Have?

In *Machines Like Me*, and even in Jonze's film *her*, there seems to be little effort to disguise the melancholia that attaches to the encounter between the human and the human adjacent. *Klara and the Sun*, Ishiguro's novel, is also suffused with a curiously affectless melancholy, a paradoxical "mood". This atmosphere of sterility and melancholy is a vision of a future in which humanoid—human adjacent—robots function as companions for human subjects whose world seems to have been depleted (which is why the human adjacents become necessary). Yet the question remains whether the human adjacents can fill the void by their companionship, whether there can be any true bridge or meaningful equality between humans and nonhumans, or whether such equality will ever be achieved, will ever underwrite a claim to equal rights between humans and nonhumans. As in *Never Let Me Go*, Ishiguro's characters in *Klara and the Sun* (Ishiguro 2021), both human and human adjacent, seem to live in a bell jar, evacuated of anything like breathable air. The setting is the not too remote future, a world that seems de-worlded, in the sense that it is peculiarly etiolated. The AIs, even the commercially available human adjacents affordable by the wealthy, are far more sophisticated, closer to human intelligence, than anything imaginable today; yet there is little else in this world that seems technologically advanced. Although McEwan's *Machines Like Me* also distorts history, he does mention "autonomous" cars as well as computer science (including especially artificial intelligence) (McEwan 2020). By contrast, Ishiguro does not have his characters dwell on technology as a topic for discussion quite as much. There are no other technologies with any significant presence—no self-driving cars, trains, or airplanes; and there is even a throwback "Cootings Machine" that spews smoke like some industrial age machine.

Klara is a solar-powered human adjacent AF, or "artificial friend", chosen by the 14-year-old but infirm Josie to be her companion, living as her human adjacent in a country house. As with Charlie Friend in McEwan's *Machines Like Me*, Josie's unnamed illness for the most part isolates her from others, with the exception of her friend Rick—and her "oblong" (an iPad-like device). Because she is at a difficult age and lonely, she seeks in Klara a helpmate, just as Charlie had sought one in Adam, his human adjacent. But just as Adam turned out to be a less than perfect companion, Klara too is not everything Josie might have wished for.

Indeed, as noted, artificial friends like Klara and other robots pose a threat, displacing human workers (including "elite workers") from their jobs, though they are themselves replaceable by other, better or more advanced, AFs. As she waits for someone to buy her from the store, Klara is pushed to the back by new and improved arrivals. To the extent that Klara is able to determine the contours of her own existence, however, it is within the constraints of her programming. As the manager of the store (where AFs like Klara are sold) puts it without mincing words, "It's for the customer to choose the AF, never the other way round" (Ishiguro 2021, part One).

Yet Klara is the one Josie chooses, because she makes a special connection with her, something she has not been able to have with other human beings. Every AF has a unique "personality", programmed to suit the particular needs of the child whose family purchases the AF as a human adjacent companion. Some AFs are more like toys or at best robots, others like Klara are indeed humanoid, approximating a real human intelligence. Gifted with powers in areas where the humans themselves fall short, Klara has a keen eye for observing details. She is able to learn a great deal about humans by their voice, body language, and gaze, even as she herself is on display for prospective buyers in the store selling artificial friends, continually producing elaborate if wrongheaded (over-)interpretations of their motivations and intentions. (Does that humanize Klara, ironically?) The manager of the AF store emphasizes Klara's observational sophistication as a selling point for Klara as a member of the B2 series of AFs sold by the store:

> Klara has so many unique qualities, we could be here all morning. But if I had to emphasize just one, well, it would have to be her appetite for observing and learning. Her ability to absorb and blend everything she sees around her is quite amazing. As a result, she now has the most sophisticated understanding of any AF in this store, B3s not excepted. (Ishiguro 2021, part One)

But Ishiguro adds a wrinkle to Klara's gift, *seeming* to confer on her a capacity for "feelings" beyond "rationality". Can she genuinely feel pain, sorrow, or joy? A great deal hangs from this question: her relative status as being able to claim legal and social rights enjoyed by human beings, and the sharpness of the divide or rift between the human adjacent and the human. As Klara says, "I believe I have many feelings. The more I observe, the more feelings become available to me." Yet Klara is—well, bloodless, affectless—as she dispenses (desperately) upbeat and hopeful observations. If Klara finds reason for what she describes as hope—even faith—that the sun will be not only a provider for her but also a protector for Josie, she does so without actually experiencing anything like hope. Her most remarkable trait is an inhuman affectlessness, a flatness that pervades this novel, draining it of humanity—but that seems to be a way of posing the question of humanity, of "being" itself, the question Heidegger in *Being and Time* sees as critical to authentic existence. Klara's sometimes misplaced and frequently naive optimism about the power of the sun's rays to ensure a desirable future for "her" human owner thus feels to the reader like a cruel joke not only about the fact that she literally takes her power from the sun but also because her sunny disposition makes a mockery of humanity's actual conditions and prospects. If this human adjacent could claim rights and aspire to personhood on the basis of "consciousness" like humans, then it would at best qualify as false consciousness. The story scuttles the hope that human beings can build a companionable world in which nonhuman AIs might claim rights and even personhood.

Klara is certainly able to ponder the consequences of her actions and attitudes, especially the issue of how her actions affect Josie, whose welfare is her chief concern. But this is not human concern. Josie herself treats Klara more or less like a fellow human being, not merely a commodity or toy. She wants Klara as her human adjacent, but as she tells Klara, only by mutual consent:

> 'You will come, right? If Mom says it's okay and everything?' I nodded encouragingly. But the uncertainty remained on [Klara's] face. 'Because I don't want you coming against your will. That wouldn't be fair. I really want you to come,

but if you said, Josie, I don't want to, then I'd say to Mom, okay, we can't have her, no way. But you want to come, right?'. (Ishiguro 2021, Part One)

Klara is vulnerable, at least in Josie's perception, to a nearly humanizing "uncertainty". She is also susceptible to a near-human anxiety, although she has no facility in expressing it as a human being would, about whether after their first encounter in the store, Josie would return to buy her, and about whether someone else would buy her before Josie could, or whether she would remain unwanted and unbought, *nobody*'s AF. The hope on which the story turns is whether Klara can be a factor in Josie's recovery from her debilitating condition; but it is also hoped that whatever the outcome for Josie, Klara will also be able to be a support for Josie's mother. Indeed, it becomes clearer as the story unfolds that it is actually the mother whose needs—including assuaging her guilt regarding the way she had intervened in Josie's life—are most to be serviced by Klara. Josie's mother had endangered Josie's health in her ambition to have Josie "lifted" into the more privileged echelons of society, like other children of the affluent. If Klara the AF is analogous to the Hegelian "slave", then it is not Josie so much as the mother who takes the position of the "master"—who is, as noted above, inextricably tied to and constitutively dependent on the slave.

Klara is capable of saying she *feels* powerless (is this a capacity for emotion, feeling, affect *tout court*?) to determine her own fate. And she is certainly able to enunciate what looks like human compassion (to feel *compassion* is surely to *feel*?), as when she encounters a beggar man and his dog on the street, who suddenly disappear one day, and she (as it turns out mistakenly) surmises that they had died. She appears gripped by sadness: "The most important thing I observed during my second time was what happened to Beggar Man and his dog . . . . I noticed Beggar Man wasn't at his usual place greeting passers-by . . . . I didn't think much about it at first because Beggar Man often wandered away, sometimes for long periods. But then once I looked over to the opposite side and realized he was there after all, and so was his dog, and that I hadn't seen them because they were lying on the ground . . . . it was obvious they had died . . . . I felt sadness then, despite it being a good thing they'd died together, holding each other and trying to help one another (Ishiguro 2021, Part One). Similarly, upon encountering a bull, Klara expresses strong "fear": "I happened to look to my left, over the fence running beside us, and saw the bull in the field, watching us carefully . . . . I was so alarmed by its appearance I gave an exclamation and came to a halt. I'd never before seen anything that gave, all at once, so many signals of anger and the wish to destroy . . . ." (Ishiguro 2021). As a range of thinkers from Søren Kierkegaard to Jacques Lacan have emphasized, to feel this particular kind of fear, of the "fight or flight" variety, is at least an animal capacity, but to feel anxiety is specifically human; I have elaborated the point elsewhere, but allude to it here to emphasize how Klara's case complicates things (Kierkegaard 2014; Lacan [2004] 2015).

Importantly, Klara also appears sensitive to the *lack* or *deficit* of feeling in humans and even in the habitus or "world" she inhabits. Following an unsuccessful "family" outing to Morgan's Falls, Klara *reflects* (suggesting consciousness) that she found "puzzling . . . . the change Morgan's Falls made to the Mother's manner. I'd believed the trip had gone well, and that there would now be a *warmer atmosphere* between us. But the Mother, just like Josie, became more distant, and if she encountered me in the hall or on the landing, she'd no longer greet me in the way she'd done before" (Ishiguro 2021, part Three). Expressions such as "puzzling" and "believed" and "a warmer atmosphere" seem to belong to a consciousness that is not just machinic. Surely a mere robot could hardly "believe", or desire "a warmer atmosphere"? Furthermore, Klara even ponders the implications of such a mood shift in her humans for herself, suggesting a *reflexivity*, a *self-consciousness* about feelings and affects:

> the possibility came into my mind that my limitations, in comparison to a B3's, had somehow made themselves obvious that day, causing both Josie and the Mother to regret the choice they'd made. If this were so, I knew my best course was to work harder than ever to be a good AF to Josie until the shadows receded.

At the same time, what was becoming clear to me was the extent to which humans, in their wish to escape loneliness, made maneuvers that were very complex and hard to fathom . . . ". ([Ishiguro 2021](), part Three)

The *trope* of an AI acceding to such a level of "consciousness" is almost (but not quite) unremarkable now. The challenge is to remain both skeptical of easy ascriptions of human consciousness and therefore legal and social rights (as membership in a community implies) while simultaneously remaining open to a paradigm shift that would make it necessary to accord rights and legal and social personhood to nonhumans. But what about Klara as the master of her own life outcomes—as someone conscious enough to be capable of claiming right, of using a rights discourse to make a contribution? Would she qualify as a legal person or a social person? What autonomy, what rights, could she claim—could she claim to be a free subject, with personal sovereignty and power of determination over her being? Does she possess (self-)consciousness, the hallmark of human personhood?

Klara may be competent as an AI—she is able to negotiate the world and plan for Josie's day-to-day success in that world—and she is very capable as a human adjacent. She expresses and demonstrates care for Josie's present and future, thus fulfilling admirably the role intended for her to be a devoted companion to her human. Yet she also reveals "intellectual" or cognitive shortcomings, most of which have to do with the fact that her intelligence is artificial, so she has to compute or recognize patterns, where human intelligences might "think" or even intuit, feel, or move instinctively. Her childish naivete is striking, especially in her behavior regarding the sun. She enjoys watching the sun set over the horizon and makes great effort to witness this daily event, even at the risk of breaking decorum and infringing on a neighbor's property, as well as potentially inviting the displeasure of Josie's mother. Though she certainly is justified in seeing the sun as what gives her her hope, her life, and her energy (she is after all solar powered and states that she is "nourished" by the sun), if she were human we would say she goes too far in her mad adorations, putting her faith in the sun's ability also to cure Josie.

It is hard to take comfort then from the positive description of the novel by the Booker Prize nominating committee: "What stays with you in *Klara and the Sun*," as they put it in longlisting the novel for the prize, "is the haunting narrative voice—a genuinely innocent, egoless perspective on the strange behavior of humans obsessed and wounded by power, status and fear." While the humans are negatively portrayed as self-serving, the robot hardly promises a richer world. So again the question of the human presents itself: Can the AI human adjacent think? Can it claim the same rights as the human? Klara as a human adjacent deserves *some* rights we might give to another human being—rights to which she would be entitled, presumably, even if she did not have (self-)consciousness. Interestingly she does seem to at least simulate (self-)consciousness when she "reflects" on her capacity for memory: "Over the last few days, some of my memories have started to overlap in curious ways. [ . . . ] I know this isn't disorientation, because if I wish to, I can always distinguish one memory from another, and place each one back in its true context. Besides, even when such composite memories come into my mind, I remain conscious of their rough borders . . . ([Ishiguro 2021](), Part Six).

By seeming to extend (self-)consciousness to an AI or human adjacent, the novel can apparently spur "anxiety" in some readers about the "future of capitalism" and about "the nature of dignity, existence, and humanity." The book "makes vivid the way in which emerging technologies developed by profit-seeking corporations are reshaping our private lives and leads us to reflect on the role that such technologies have and should have in our social environment" ([Majia and Nikolaidis 2022](), esp. p. 303). An important facet of the anxiety produced by the machine-human (Machihuman?) adjacent is that it is *human-displacing*—not only displacing human workers from their jobs (including, in Ishiguro's novel, elite jobs), but provoking also a deeper, epistemological anxiety about what remains the province of the human. We see evidence of such human displacement in Jonze's film, as well as in the two novels by McEwan and Ishiguro.

This is why I raise the issue: when sophisticated humanoids or human adjacents such as Samantha (in *her*), Adam (in *Machines Like Me*), and Klara (in *Klara and the Sun*) enter into a habitus shared with humans, disturbing the presumption of human superiority and self-sufficiency, what is left of the human, and, what rights can the human adjacent claim? A secondary issue is: What rights might be called human *adjacent* rights, extending the category for instance not only to pets but arguably all nonhuman animals as well as the anthropomorphized things in Kondo's books and show? Such issues are becoming even more vexed in our contemporary era in which we are witnessing the astonishingly rapid advances in AI such as Chat GPT, when artificial intelligence may no longer be modeled on *human* intelligence, and may become a truly emergent alien intelligence.

## 6. Part V: The Eclipse of the (Black) Human, or, Afropessimism as Technology of Dehumanization

The rise of things as actors in the world shared with humans, robots, disembodied AIs, and nonhuman adjacents with bodies, presents the challenge of an ("imaginary") eclipse of the human as received category. This is a bifurcated discourse of antihumanism, produced by the new technologies of the human. It adds a new turn on the question of what is "left" of the human in the wake of such a discourse of antihumanism (or posthumanism), and what kind of *distribution of rights* might be entailed by such antihumanism—for a discourse of antihumanism is also an expression of a desire to rethink and reframe the category of the human. I want to suggest then that today we are seeing two vectors of this discourse of antihumanism:

(1) A general, epistemological discourse that takes into consideration the waning of the presumed exceptionalism and sovereignty traditionally claimed by or accorded to the human—to the anthropocentric worlding of the world. This consideration presents a critical perspective on a range of registers. On one end of the discursive spectrum is an environmentalist or "green" critique of the damage wrought on the planet by the presumption coded into the customary anthropocentric worlding of the world—as if the planet were meant to service and support human life as privileged over all other entities, including "things" (a critique presented for example in speculative realism and in OOO, both mentioned above—as in the work of Quentin Meillasoux and others) ([Meillasoux [2008] 2011]). At the opposite end of this range of discourses is an epistemologically oriented question also introduced above: What is left of the human in a worlding of the world in which things, robots, AIs, and (non-)human adjacents are significant actors, *co-producing* the world shared with humans? There has also been much discussion of the relations among things, independent of human intervention or action—as in the work of Jane Bennett.[8] This epistemological discourse is beyond the scope of the present essay.

(2) There is another, ontological discourse that I develop in this section of the essay: a discourse of antihumanism that finds its culmination today in Afropessimist theory and other work. This intellectual corpus challenges the very ontological construction of (black) humans, and in doing so also puts into question white humans and by extension the very category of the human as it is now understood: as requiring the abjection of some nonhuman animal, some thing, some gendered or racialized other. The premise here is that the human world, as it is worlded at the present conjuncture, inheriting the long history of "civilization", requires this procedure of abjection. In Afropessimist critique, the black man is a special, absolute, case of this abjection. The "*différance*" with the human adjacent AI is therefore instructive, given the problematic discussed earlier in this essay.

Afropessimism confronts us then with a deconstruction of the whole discourse of the human, of technologies of the human—a problem for EuroAmerican thought itself. It emphasizes an inassimilable surplus, the excess nominated "blackness", as symptom of whiteness ([Dayal 2012]). This inassimilable excess is structurally, "katechontically", excluded from the discursive space of the human ([Kaplan 2019], esp. pp. 69, 80). On this

reckoning, blackness is always already a figure for what Calvin Warren terms *onticide*: it "sustains the very fields of existence" (Warren 2017, see esp. p. 392).

Sustaining the fields of existence is a metonym for the project of *worlding*. The project of worlding involves constructing the (normative) category of "Man" from which blackness must be *constitutively* excommunicated, abjected. Simultaneously, whiteness (or normative, transparent, "Man") is *ex-nominated* in this procedure, without acknowledging the exnominative exclusion, which is the mirror image of the exclusion of the black man from the category of the human (at least from the perspective of Afropessimism). Consequently, blackness can only be invoked *sous rature* (under erasure), as an impossible yet fungible category of the defining symptom of the normative. This, I would argue, is not an "erasure" reserved only for the black (man), but also applies to the subaltern and other abjected figures—abjection being crucial in the technics of the human, for this primordial exclusion from what are defined as "human rights" cannot encompass an account of the a priori exclusion. The primordially excluded abject is katechontically foreclosed from entry into the shared space of the human. If it were to present itself, we would encounter the Real, the apocalypse of the human world. But for the moment, let us leave aside this more general *techné* of the erasure of subaltern and other "others", given that the immediate focus is on Afropessimism.

Well before the rise of Afropessimism as a critical discourse, Frantz Fanon was calling for what I have elsewhere described as an ethical antihumanism, a deconstruction of humanism as a system of thought, as epistemology of "white" and "black", but also of "brown" (Dayal 2008). In *The Wretched of the Earth*, Fanon exhorts Europe and humanity urgently to inaugurate a revolution: "comrades, we must grow new skin, we must work out new concepts, and try to set afoot a new man" (Fanon 1963). This re-epidermalization will not be, if it succeeds, just skin deep, it will not even be artificial flesh, for that would be predicated on the real flesh of the human—in the way that the creation of Eve reinstated Adam's rib, his flesh, as the matrix out of which she had emerged, as if expropriating the woman's capacity for giving birth from her own real flesh. Rather it will be a deconstruction, an excarnation, to adapt Richard Kearney's phrase, of the fleshed human, the human as actually existing (Kearney 2021). In that regard, Fanon asymptotically approximates, indeed anticipates, Afropessimism. Afropessimism seems to take a page from Fanon's diagnosis that the black man inhibits only "a zone of non-being", what Alain Badiou describes as "inexistent existence." (Badiou 2011). Thus, the double-bind, Warren acknowledges, is that Afropessimism encounters the quandary simultaneously of accounting for the primordial exclusion of black "man"—not to mention black "queer" or black "woman"—from the structuring economy of difference that constitutes being within the violent hierarchy of humans over but dependent on nonhumans, *and* accounting for those various "differences" and particularities. "[S]ince the black is *not* a human," declares Warren, "it cannot claim 'difference' or 'particularity' as a feature of existence (because these belong to the human)." (Warren 2017, p. 394. Emphasis original). In that regard, Fanon asymptotically approximates, and anticipates, Afropessimism.

As human adjacent, however, the AI, automaton, or robot also poses a set of challenges, which are fundamental provocations: (1) what is the ontological status of these adjuncts to (or exclusions from) the human? (2) What social and legal rights might they command or demand? And (3) what does it mean to afford or deny the human adjacent supplement (the abjected excess, the (non-)subject or nonhuman) equal footing with the human under the aegis, and thus legal protection, of "human rights"? Would such affordance of equal footing not also render human rights themselves inoperable in principle, for there would be no special legal standing given to "human" under human rights?

Minimally, human rights confer certain inalienable guarantees to humans but not other beings or things in the world. This is today increasingly being interrogated as an unwarranted exceptionalism. Appositely, Wendy Brown notes that there is a criticism of human rights discourse as among other things "entailing secular idolatry of the human."

This criticism is taken seriously by Michael Ignatieff; but ultimately he takes a position that Brown herself does not endorse ([Brown 2004](#)).

Still, taking this criticism of human rights discourse as idolatry of the human seriously means at the very least acknowledging the nonhuman or human adjacent as also having agency or having the status of an actor or actant in the sphere of thought and action that human beings themselves inhabit, as Bruno Latour suggests in developing his actor network theory or ANT ([Latour 2005](#)). Indeed, it means also acknowledging that these nonhuman actants are also instrumental in *making*—co-producing—our shared world. It also entails adopting a new perspective on the status of things as things.

The contemporary moment, witness to the burgeoning of new technologies of the human and the (non)human adjacent, such as ChatGPT, has only exacerbated the condition diagnosed by Adorno and Horkheimer, that in the modern, post-Enlightenment era of mass production, increasingly the human is at risk of reification, if not self-alienation to the point of volatilization: "The countless agencies of mass production and its culture impress standardized behavior on the individual as the only natural, decent, and rational one. *Individuals define themselves now only as things*, statistical elements, successes or failures" ([Horkheimer and Adorno 2002](#)). If individuals define themselves now only as things, nonhuman entities seem to fill the vacuum opened up by the vacation of the human from the habitus of the individual. Artificial intelligence may very well soon supercede what human intelligence could even imagine or conceptualize as intelligence or even consciousness. We "might find ourselves living inside the dreams of an alien intelligence", as Yuval Noah Harari recently argued in *The Economist* ([Harari 2023](#)).

Things as supplements or "excessive" accoutrements (things as abjected "blackness" for instance) compel a rethinking of the human, and, metonymically, human rights. These nonhumans or human adjacents emerge in these cultural productions as actors within the sphere of human thought and action. They do not so much "ventriloquize" us as enable a projection of us as humans, a projection that opens up a virtual space for a new technology of the human that also encompasses the nonhuman. This technics of the human forces open a question about what remains human in spaces designated "human", the question, as Agamben might put it, of the *use* of a not-yet-gendered or -raced body, as well as the question of the status of "human rights": whether those human rights are extensible to nonhumans or human-adjacents. (See ([Agamben 2016](#)). See also ([Agamben 2002](#), p. 41)). In the case of this supplement or excess—the human adjacent agent in the human "world"—the figure of the human risks becoming mere catachresis or mystification of an entity that is itself no more than a thing.

Yet this technics is also now unmasked as a *tuché*, a missed encounter, as Jacques Lacan theorizes it, with humanity as such. In his *Tuché and Automaton*, which apostrophizes Aristotle in the very title, it is a similar problematic that emerges. For Lacan, the subject is relatively determinate as a product of mechanistic technics or law. At a first level, the subject is construed as *automaton*. Here, the subject attaches itself to a sliding chain of signifiers (s1, s2, s3 . . . .), each of which ultimately proves not to be coherent and self-sufficient; it is not vouchsafed sovereignty. Thus, the subject remains always "barred", without ever arriving at a stoppage of the sliding chain of signifiers, at total satisfaction of (unconscious) desire. It is at a second level that Lacan situates the *tuché*. This is the level of the missed encounter with the Real, where the human could be re-engineered or where a new technics of the human might become possible. But why is this the case? It is because this reconfiguration of the human can only happen in a post-deconstructive moment, that is only when the sliding chain of signifiers is rendered inoperative, stops "writing itself", in Lacan's terms ([Lacan 1978](#)).

This is the kind of post-apocalyptic, post-katechontic, moment figured in Afropessimism, but prefigured in Fanon's discourse of the "new Man", a projection that can rise only "in the wake" (to adapt Christina Sharpe's image) of the deconstruction of black man as well as white man. And to imagine with Fanon the "wake" of the deconstruction of "race", as Frank Wilderson suggests, is to recognize the impossibility of such a dismantling

of the regnant epistemological structure: hence the pessimism structuring "Afropessimism" itself. Furthermore, the deconstruction of race would have to be conducted intersectionally, along with a deconstruction also of gender, as Hortense Spillers suggested in "Mama's Baby, Papa's Maybe: An American Grammar Book" (Spillers 1987).

In such a post-apocalyptic, messianic future anterior, a new horizon for human rights may become possible, may indeed be required. *But we do not know what that means yet*. What would the "human" in human rights mean in that messianic future, when a new worlding of the world will be necessary, even inevitable? Here, I suggest, Heidegger's notion of the rift or *Riss* between *world* and *earth* is illuminating. This *Riss* is simultaneously conceptual design and the abyss/breach between the phenomenal world that emerges in the "clearing" and the earth—as the groundless ground from which "world" emerges or by which it is disclosed (Heidegger [1950] 2002). And it is as if Heidegger here anticipates the Lacanian abyss that divides the real, that which is actual, from the Real concealed by the fabric of the real. (The Real is one of the three registers that Lacan theorizes in his discussion of the Borromean knots—the Symbolic, the Imaginary, the Real; later, Lacan would add the register of the *Sinthome*). On this view, Heidegger would have understood Lacan's linkages and separations of the Symbolic from the Real in terms of the dialectical rift between world and earth. The real or "worlded" world of the human is just the textile of "design", or *Riss*, in Heidegger's lexicon. Both the human symbolic and the imaginary registers seek to position a screen or defense against the threatening Real, though sometimes the screen tears or permits the Real to extrude: this is a moment of trauma because it shatters the smooth surface of the real world. As indicated, for Heidegger, the opposite term for "world" is "earth" (Heidegger 1993). World in this sense is made actual in the present scene of humans alongside things, nonhumans, and human adjacents—all of whom can be actors within the shared world they have worlded together.

In *The Open*, Agamben asks, "What is man, if he is always the place—and, at the same time, the result—of ceaseless divisions and caesurae? It is more urgent to work on these divisions, to ask in what way—within man—has man been separated from non-man, and the animal from the human, than it is to take positions on the great issues, on so-called human rights and values" (Agamben [2002] 2004). The open is the interval within which the question of the human must arise: "To define the human not through any *nota characteristica*, but rather through [ ] self-knowledge, means that man is the being which recognizes itself as such, that man is the animal that must recognize itself as human to be human." (Agamben [2002] 2004, p. 26).

While Agamben's analytic emphasizes the abyssal or division between the human and the animal, he does open the analytic frame when he maintains that the "anthropological machine of humanism" constitutes an "[i]ronic apparatus" that does not hypostasize a "nature proper to *Homo*", but rather "verifies its absence". It "suspend[s]" *Homo* "between animal and human—and thus, his [sic] being always less and more than himself." (Agamben 2016, p. 29). What kind of "anthropological machine" is this? Agamben specifies that "[i]n so far as the production of man through the opposition man/animal, human/inhuman, is at stake here, the machine necessarily functions by means of an exclusion (which is also already a capturing) and an inclusion (which is always already an exclusion)." (Agamben 2016, p. 37). There is, however, also an absolute exclusion of what Agamben maintains is "bare life", against which the human is produced: "neither an animal life nor a human life, but only a life that is excluded from itself—only a *bare life*." (Agamben 2016, p. 38). Bare life poses precisely the question of human rights in the strict sense of *rights proper to the human*. For bare life is that life which is excluded from all legal, moral, social, or political ecologies: *bare life is radically de-worlded, denied worlding*. Bare life can thus be destroyed with absolute impunity.

So what recourse is available, asks Agamben: "faced with this extreme figure of the human and the inhuman, it is not so much a matter of asking which of the two machines (or of the two variants of the same machine [here Agamben means the ancient variant and the modern]) is better or more effective—or, rather, less lethal and bloody—as it is

of understanding how they work so that we might, eventually, be able to stop them." (Agamben 2016, p. 38). This may be a fond hope presented in the anterior future tense: it presumes that in the future the dialectic of othering between human and nonhuman, between master and slave, between self and other will have been rendered inoperative, in an impossibly quiescent synthesis. Does Agamben not himself imagine this synthesis when he envisions the "total humanization of the animal coincid[ing] with a total animalization of man." (Agamben 2016, p. 77).

The question I want to pose, however, is what happens when the borders between the human and the nonhuman become porous and blurred. We need a more generalized regime of rights for nonhumans, including things (as well as animals), on the model of the Nonhuman Rights Project, whose website suggests that "[a]s with human rights, nonhuman rights are based on fundamental values and principles of justice such as liberty, autonomy, equality, and fairness. All of human history shows that the only way to truly protect human beings' fundamental interests is to recognize their rights. It's no different for nonhuman animals" (The Nonhuman Rights Project 2023). The project focuses on nonhuman animals—but as I am suggesting in this essay, something far more sweeping may ultimately need to be contemplated.

What kinds of rights would become necessary to re-world the world if we desire a justice that is extended to all nonhumans, beyond just nonhuman animals? That is, to *things* as agents, to robots, and especially to AIs, including human adjacents. Of course, these nonhuman actors or agents cannot all be homogenized into a generalized otherness such as nonhumanness. Still, the challenge is to imagine re-defining the human, reimagining the *techné* of the human, without necessitating that bare life be de-worlded, or without conceding too much to Heidegger's related "triple thesis", which Agamben himself invokes, that "the stone is worldless [*weltlos*]; the animal is poor in world [*weltarm*]; man is world-forming [*weltbild*]"? The worldless stone (or, in my terms, the thing) is "quickly set aside" by Heidegger, Agamben notes, when Heidegger sets out on his "inquiry with the middle thesis, immediately taking on the problem of what it means to say 'poverty in world.'" (Agamben [2002] 2004, p. 51). The ontological status of the environment is specified as open (*offen*) but not disconcealed, openable (*offenbar*). For the animal, beings are open but not accessible: they are open in an inaccessibility and an opacity—that is, in some way, in a nonrelation. This *openness without disconcealment* distinguishes the animal's poverty in world from the world-forming which characterizes man." (Agamben [2002] 2004, p. 55). I am arguing that today we need something more nuanced, accommodating the human adjacent without conceding too much to it.

By extension, if the poverty of world is what separates the human from the nonhuman animal, it is what divides the human from the nonhuman robot AI or human adjacent. We could say it is analogous to "consciousness". Human beings have it, AIs do not. And it is here we might take up again the question of what rights might accrue to this othered group. Even in Heidegger we can glimpse a possible pathway: to *be able to ask the question* of becoming poor in world or de-worlded is the condition of being human: this is a radical self-reflexivity, consciousness properly speaking, in that it can entertain, as an animal cannot, being de-worlded, the question of the possibility of being deprived of the "disinhibitor" (*das Enthemmende*), the set of conditions or the environment in which a human can world his or her world. (Agamben [2002] 2004, p. 51). It is another form of posing the very question of being. It is a radical openness.

Agamben returns us to Heidegger's theorization of a rift (Heidegger's term in his *Aesthetics* is *Riss* but Agamben notes that Heidegger frames the "reciprocal opposition of world and earth" as strife, *Streit*) between world and earth. (Agamben [2002] 2004, p. 72). Glossing this rift or strife, Agamben explains that if "in the work the world represents the open, then the earth names 'that which essentially closes itself in itself.'" Agamben goes on to quote Heidegger's account, "'The Earth appears only where it is guarded and preserved as the essentially Undisclosable, which withdraws from every opening and constantly keeps itself closed.'" What then is the contribution of the work of art, what is the work of

representation? Agamben offers the answer that "[i]n the work of art, this Undisclosable comes to light as such. 'The work moves the earth itself into the open of a world and keeps it there.' 'To produce the earth means: to bring it into the open as that which closes itself in itself [*In-sé-chiudentesi, Sichverschließende*].'" (Agamben [2002] 2004, p. 71).

It is possible, as my remarks above suggest, to extend Agamben's point about the "anthropological machine" as opening up the interval or "suspension" of *Homo.* The interval opens *also* being between the human and *all* nonhuman others, not just animals but also things, humanoid robots, AIs, and human adjacents. It is not just via a recognition of the animal within us that we re-cognize the human—the "animal that therefore I am," as Jacques Derrida put it in his book of the same name (Derrida 2008). It is also via the relation or world co-production shared by human and the nonhuman that the human itself can be put into question. That indeed is the "open". The re-cognition of the human as human is routed through an encounter with an intimate other, in the space of the open. The re-cognition depends on a second recognition of the human as newly unfamiliar, in a reflection back to the human. This is an "undisclosure" of "his" being (to use the Heideggerian idiom) in an act of splitting and doubling of the image of the human in a form that is both asymptotic and excessive, mimesis and extimacy. "Extimacy" because the nonhuman is also humanoid without subsuming or being subsumed by the human—both radically different and yet intimate. *Mon semblable, mon frère*, to quote the Baudelairean formula that captures this extimacy: "an exclusion (which is also already a capturing) and an inclusion (which is always already an exclusion)," in Agamben's own words, quoted above.

## 7. Conclusions

As the line of argument presented above indicates, AIs should be afforded *social* rights, but not *human* rights—this is the point I made about Ishiguro's human adjacent Klara, but it applies also to the other human adjacents discussed, including McEwan's "first man", Adam, an AI, and to the human adjacent voiced by Scarlett Johansson in the film *her*—as well as to the things and everyday household commodities that Marie Kondo lavishes attention on as if they completed the human world and were therefore in a sense human adjacent as well. Failing to grant some rights, to AIs especially, can be said to be wrong simply because to deny AIs respect is to deny respect to *human* intellectual activity and ingenuity (if not to AIs as intellectual *property*). It is to fail to afford respect to ourselves, ultimately. Among these minimal rights might be the right of AIs to not be destroyed or disposed of irresponsibly. There is a moment in *Machines Like Me* when Charlie seeks to destroy Adam with a hammer. This is desperate, even unjust, if we believe that we owe or in the near future will owe AIs and robots certain moral considerations—minimal rights. Thus, Visa Kurki can imagine a Robot Welfare Act to "prohibit[ ] certain particularly gratuitous acts of cruelty toward robots " (Kurki 2019). Another right AIs might claim is the right not to be used as proxies in activities we might see as immoral for human agents, and the right not to be used to absolve humans themselves of racist bias, as when algorithms are used to allow their human users to enact discriminatory actions against racial, ethnic, sexual, or other minorities—or as when an AI is used to screen out applicants for jobs in a way that human beings might consciously want to disavow (e.g., denying jobs to such minorities by using algorithms to detect non-mainstream identity markers that human eyes might not have been able to detect, or algorithms that a capable human reviewer of algorithms might have disqualified as racially discriminatory).

Granting human adjacents, but most particularly AIs and robots, some of these social rights is different from granting them social personhood, just as granting AIs and robots legal rights is different from granting them legal personhood. (Kurki 2019, p. 178). At this historical juncture it would be reasonable to maintain the distinction, affording some limited social rights as well as some legal rights to AIs, especially what I have identified as human adjacent AIs, while denying them full human rights as well as social personhood. This remains true for animals as well. We risk an epistemologically messy moral landscape if we do not protect human rights as well as social personhood (to say nothing about legal

personhood) exclusively for "wet brained" human beings. This does not condemn us to a vicious humanist exceptionalism. I would agree then with Kurki that "[a]nimals hold, and slaves held, similar legal rights, yet animals and slaves are widely—and correctly—classified as legal nonpersons. Thus, we need to distinguish robots as right- holders from robots as legal persons." (Kurki 2019, p. 178). Of course I would not support slavery itself in any context. But in this context, it is not simply a matter of "doing the right thing", to risk a pun. There would be a significant cost to extending fundamental human rights to humanoid robots, AIs, and human adjacents. Kurki rightly points out that "humanoid robots would likely qualify as legal persons if they could no longer be owned and if they received wide-ranging fundamental protections (for instance, attempts to shut them down would be classified as attempted homicides)." At the very least, this misguidedly magnanimous gesture would create havoc in the legal and social order. It is better to maintain the axiological "is/ought" distinction: whether robots can be afforded rights is an "is" question, rather than an "ought" question, as David Gunkel suggests (Gunkel 2017, esp. pp. 88–90).

Human rights should probably be reserved for wet brained human creatures—at least as those human rights are codified even in the very first clause of the preamble of the Universal Declaration of Human Rights: "recognition of the inherent dignity and of the equal and inalienable rights of all members of the human family is the foundation of freedom, justice, and peace in the world . . . ." Likewise, the first Article of the Declaration states in its memorably lapidary style, "All human beings are born free and equal in dignity and rights" (The United Nations 1948). At least minimally, the feature of being *born* free seems to require that while some social and legal rights are due to AIs and human adjacents, given the rapid developments in artifical intelligence capacity, they must at least at this conjunctural moment, and for the sake of avoiding epistemological confusion, be denied social and legal personhood on such "biological" grounds, even if only to avoid a totalitarian biopolitical regime. By the same token, things may be recognized as potential actants in a given assemblage, any "worlding of the world", but they should be denied agency, which is premised on personhood. Yet this is a deliberately "open" orientation, rather than an attempt to impose a universal order, even a universal regime of rights, onto all humans and nonhumans. My recommendation then hews to a species of what Agamben theorizes, following Heidegger, as the open: a kind of *Gelassenheit* or radical tolerance—a live and let live orientation—towards the nonhuman as adjacent to the human, or as a companionate agent in the world that is worlded by the human being:

> Thus, the supreme category of Heidegger's ontology is stated: letting be. In this project, man makes himself free for the possible, and in delivering himself over to it, lets the world and beings be as such. However, if our reading has hit the mark, if man can open a world and free a possibile only because, in the experience of boredom, he is able to suspend and deactivate the animal relationship with the disinhibitor, if at the center of the open lies the undisconcealedness of the animal, then at this point we must ask: what becomes of this relationship? In what way can man let the animal, upon whose suspension the world is held open, be?

Similarly, my goal here has been to suggest that what is interesting and worth reflecting on is how thin is the line separating these epistemological categories. Let it stand. Let it be, for now.

**Funding:** This research received no external funding.

**Conflicts of Interest:** The author declares no conflict of interest.

## Notes

1   (Dayal 2019, see also my chapter on "Normative Microcosmic Cosmopolitanism" in this book).
2   Kondo's work is not valorized here but rather critiqued for its support of consumer culture. Even literary critics cannot ignore the fact that her work, with its enormous popular appeal, can be read contrapuntally, as posing analogous questions about the

relationship between the human and the nonhuman. Precisely because Kondo's work is not "high culture", and may arguably be an example of degraded bourgeois false consciousness, we can read it as a document of culture that reveals important cultural values, tastes, and preferences. If we are interested in the function of the human adjacent in literary works such as McEwan's or Ishiguro's, we cannot ignore that Kondo's consumerist and cloyingly "cute" (to invoke Sianne Ngai's aesthetic category) goods, toys, socks, clothing are also can be read against the grain as evidence that the issues central to this essay have cultural currency and significance, and have already infiltrated the popular consciousness. They are not dry philopsophical abstractions. Furthermore, inclusion of this popular cultural phenomenon (alongside literary works and film) to elaborate the important problematic of the relationship between the human and the nonhuman or human adjacent is also at some level a way of interrogating the privileging of high culture literary works, a way of challenging the generic privileging of literature over film and popular culture, even a way of subverting genre snobbery.

3    Huang writes, "Through this view, Kondo performs hospitable Japanese femininity through her appearance, gestures, and her choice to speak mostly in Japanese on Tidying Up. Kondo's wardrobe on the show consists of a cream blouse, cardigan, colorful skirt, black tights, and ballet flats, attire that signals respectable feminine gender presentation while deemphasizing the potential potency of female sexuality. Coupled with her petite frame, at four feet, seven inches, Kondo's performance of Japanese femininity can be understood through aesthetics of cuteness and hospitality, as someone whose foreignness not only can be welcomed into the private sphere of the home without threat but also can help one get closer to one's home" (Huang 2020, p. 1374).

4    Flisfeder and Burnham write, "it is no wonder that Fisher characterizes the present around the idea of depressive hedonia. Depression, he writes, "is usually characterized as a state of anhedonia, but [depressive hedonia] is constituted not by an inability to get pleasure so much as it is by an inability to do anything else except pursue pleasure." In postmodern consumer society, we are interpellated as subjects of pleasures that are satisfiable through objects. But because nonsatisfaction is a condition of perpetual consumption, consumer society is one that is much more productive of nonsatisfaction and a lack of enjoyment." See (Flisfeder and Burnham 2017).

5    Khamis highlights the contradiction noted above: "De-cluttering makes ironic incursions into a discourse of restraint that, in turn, informs and underpins a discourse of *alternative* consumption. See (Khamis 2019, esp. pp. 528, 514). Emphasis added.

6    The *Stanford Encyclopedia of Philosophy* notes the "widely shared insight . . . that emotions have components," and in the case of a complex emotion such as the fear experienced when one encounters a grizzly bear while hiking, these components might include an evaluative appraisal of the threat, but also a *physiological* component such as elevated heart rate, an *expressive* component (a bodily gesture or reflex), a *behavioral* component (such as the flight response), and a mental component (such as focused attention). Fundamental questions that must be entertained by any meaningful approach to understanding emotions include the issues of differentiation (how to distinguish emotions from other emotions and from non-emotions); motivation (whether or how emotions drive action and behavior); intentionality (whether "emotions have object-directedness," and if so whether one can say are "appropriate or inappropriate to their objects[ ]"); and phenomenology (whether emotions are subjective, and if so what that might mean). Although the modern conception of emotion emerged in the seventeenth and eighteenth century, and then became crucial in the Enlightenment, particularly as the foil to reason, the earliest Western traditions that focused on emotions were those of the ancient Greeks, for whom emotions were grouped, as the SEP entry notes, under categories such as "*passion, sentiment, affection, affect, disturbance, movement, perturbation, upheaval*, or *appetite*. See "Emotion", *Stanford Encyclopedia of Philosophy*.

7    It is acknowledged freely, writes Brennan, "that individualism is a historical and cultural product, the idea that affective self-containment is also a production is resisted. It is all very well to think that the ideas or thoughts a given subject has are socially constructed, dependent on cultures, times, and social groups within them. Indeed, after Karl Marx, Karl Mannheim, Michel Foucault, and any social thinker worthy of the epithet "social," it is difficult to think anything else." Yet what is interesting is that although "we accept with comparatively ready acquiescence that our thoughts are not entirely independent, we are, nonetheless, peculiarly resistant to the idea that our emotions are not altogether our own. The fact is that the taken-for-grantedness of the emotionally contained subject is a residual bastion of Eurocentrism incritical thinking, the last outpost of the subject's belief in the superiority of its own world view over that of other cultures." (Brennan 2004, p. 2).

8    See (Bennett 2010). Bennett seeks to elaborate "a more horizontal relation between human and nonhuman actants"—making more porous the divisions between them. She also seeks to distribute agency in any event, in any world-making (in my terms). Nonhuman agency then is key to any assemblage or (to return to Latour's lexicon) any network of actors.

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
