# Peer review of "Artificial Flesh: Rights and New Technologies of the Human in Contemporary Cultural Texts"

_2410-9789, doi:10.3390/literature3020018_

Round 1

Reviewer 1 Report

The is clear evidence of extensive scholarship in the conceptual fields of posthumanism, new materialism and afrofuturism and the writer has clearly mobilised this material appropriately. The primary texts referred to are thoroughly discussed and the argument clearly establishes their relevance. As such, I find that there is significant potential in the material presented and feel that, with further substantial editing, it could be resubmitted. As it stands, however, there are significant problems with its coherence. Although I realise that there are no word limits as such for this journal, these feels far too long (nearly 20000 words?), mainly because there are too many unnecessary digressions into ideas and debates which are not intrinsic to the argument (Heidegger, Lacan, Kant and so on), or if they are, relevance is not firmly established. This conceptual work needs to be more nimble and economical, as does the textual analysis, which is also prolonged and often descriptive of the plot events and characters when it needs to be more selective, synoptic and analytic. It is in these long passages on the texts when the argument, rather than coming into focus, recedes out of sight. There are some places (highlighted in my comments above) where the argument is brought sharply back into view. These questions need to be lifted forward so that they drive the analysis in advance, setting the framework for the close textual reading.

The texts used as exemplars are quite diverse - literature, film and popular culture (Marie Kondo) so I felt some further rationale for the selection would be helpful. This is especially noticeable on p.4 when Marie Kondo is suddenly brought in without much introduction or explanation and the argument appears to take a very different direction - to things more broadly rather than AI etc. I felt that the analysis of her and the two novels was the most successful and wondered whether the Marie Kondo material might work better in a separate article.

p.1 

chez Immanuel Kant?

A more subtle way of referring to the other work?

Yet there is another even more fundamental domain

We are enjoined? to ask whether a human an

P.2

Quickly spell out why lower case is significant

Two paragraph synopsis of plot is too long - abbreviate?

P.3

self-alienation of man from himself -  re-phrase?

I'm starting to notice frequent use of 'perhaps', 'suggests' 'seems' 'I would argue' and other weakening phrases. Could you review and remove where possible?

p.6 is 'thing theory' new? 20 years old?

p.7 'in a word' shd be 'in short'

p.7 what does it mean that the Falklands war was not 'disastrous' in reality? For whom? Requires qualification.

pp.8-10 very prolonged discussion account of Machines Like Me -could be condensed by removing excessive plot detail. Some paragraphs provide plot summary but no overview so it's hard to see the objective.

p. 10 should be 'one of the monster's chief complaints'? 

pp.11-14 again, there is too much summary of the plot here with the result that the overview os drowned out. Edit to condense this and bring the argument overview to the fore.

p.14 para 'This is why ...' It would be more useful if this came much earlier so that these key questions frame and direct the textual analysis.

'Yet another....' This makes the point sound like a random addition and the argument then goes in quite a different direction via Adorno and Horkheimer. I'm not sure this section is intrinsic to the argument. 

p.15 Could you consider the relationship between the two options? It feels necessary to reflect on what is at stake within each. 

'then' x 2 - rhetorically necessary?

p.16 para 1. Is it worth introducing, only then to leave aside? Remove?

Is the Fanon section really necessary? Might it be enough simply to note Fanon's anticipation of ethical antihumanism?

p.17/18. again the theoretical aside to Lacan doesn't feed into the overview so feels like another diversion.

p.18 The question about the porous is important but the argument needs to be driven more closely and consistently by these questions throughout.

p.20 'Extimacy because  .... ' needs a main verb clause.

Author Response

Please see the attached response.

Reviewer 2 Report

Thank you for this very interesting article. To further improve it, I suggest the following:

·        I think you can critically go through the whole text again with some distance and compress/shorten/clear it

·        All in all it tends to be a bit repetitive or even redundant in the examples

·        Now and then you mention theories and concepts without giving the according references, e.g. on page 1: “In this context, the discussion involves not only the issue of “cosmopolitan right,” as chez Immanuel Kant, and “cosmopolitics” as theorized by Isabelle Stengers and others but also the question of the Universal Declaration of Human Rights.” Or on page 8 “In any case the human adjacent is predictably embroiled in a version of what Hegel had discussed and Marx had developed as the master-slave dialectic; but the human, presumably occupying the position of the master in the dialectic, is also captured in the contest.” I suggest to add the references in these and other examples.

·        At the end of the introduction, I suggest to explain/describe what follows in the upcoming sections, and more importantly, what cultural products were selected and why these examples were selected. Why Marie Kondo goes together with science-fiction is not self-explanatory. I also suggest one or two sentences about the methods. Maybe a bit of the overall aim of the text as given in the abstract could enter the introduction

·        I don’t fully understand the large space given to Marie Kondo; I see that her approach is worth referring to, however, I can’t see why an organizing consultant has authority in a discussion on granting or not granting Ais social/human rights. I suggest that you clarify/elaborate more how and why her approach is so relevant for your argument, or shorten when necessary. And on page 4, I guess you could delete the text in brackets, it doesn’t really seem relevant. (The name is a knowing avoidance of the more obvious neologism she could have chosen, MariKon, unfortunately suggesting the unflattering Spanish slang term “maricón”).

·        I think the conclusion is quite remarkable: “AIs should be afforded social rights, but not human rights.  Failing to grant some rights to AIs can be said to be wrong simply because to deny AIs respect is to deny respect to human intellectual activity and ingenuity (if not to AIs as intellectual property).  It is to fail to afford respect to ourselves, ultimately.” I suggest that you summarize briefly, how from each chapter you come to that conclusion. And maybe look again into the chapters from the perspective of the conclusion and make the argument more stringent and concise.

·        Maybe this literature might be interesting for you:

o   https://link.springer.com/article/10.1007/s13347-020-00415-6

o   https://link.springer.com/article/10.1007/s00146-021-01299-6

o   https://books.google.de/books/about/Robot_Ecology_and_the_Science_Fiction_Fi.html?id=GnpwCwAAQBAJ&redir_esc=y

o   https://books.google.de/books?hl=en&lr=&id=jBSo8LIDowoC&oi=fnd&pg=PP8&ots=dPCk_fSSVe&sig=mt3IUTZh286acwRwDJ1-55I1erc&redir_esc=y#v=onepage&q&f=false – chapter 5

Author Response

Please see the attached response: my responses are in red, to distinguish them from the reviewer's text.

Round 2

Reviewer 1 Report

This is a much improved submission. Although I would suggest that it is still too long, it doesn't feel so overlong because it it is more coherent and less fragmented than the first submission. The introduction has been strengthened by a tighter and better signposted rationale and the ongoing argument takes care to explain connections and implications as it progresses. Thus, the contribution to knowledge is clearer and, as such, this article constitutes a more detailed and thoughtful analysis of the concepts and texts involved.  I would suggest a final edit to shorten the essay. There are some very long quotations, for example, and the discussion of Kondo feels over-extended at times.

Author Response

The essay's length, as this reviewer admits, no longer "feel so overlong because it it is more coherent and less fragmented than the first submission."  I thank the readers' suggestions, and as you can see from the tracked changes, have worked hard to shorten and trim where I could but more importantly I have tried hard to adjust in response what I believe are really the more important issues.  I have indeed worked line by line to be more coherent. I have tried to allay any sense of fragmentation by much more frequently signalling the connections among my various points.  I admit the essay is challenging to follow, but the main line of argument is very clear and precise. The reviewer also notes that my introduction "has been strengthened by a tighter and better signposted rationale and the ongoing argument takes care to explain connections and implications as it progresses. Thus, the contribution to knowledge is clearer and, as such, this article constitutes a more detailed and thoughtful analysis of the concepts and texts involved."  I appreciate the reviewer's generosity in acknowledging my efforts to do precisely such revision. And I have in this final edit also heeded the reviewer's exhortation to attempt to do some small things to "shorten the essay." Again, given the scope of the argument, there are costs as well as benefits to the very changes we are talking about. If I shorten too much, the essay becomes incoherent and fragment.  If I make connections more explicit, more regularly the essay grows longer.  I have taken to heart the reviewer's suggestions and shortened some long quotations, and slightly trimmed my discussion of Kondo.  If I don't elaborate on why I am taking seriously a lifestyle coach's popular work cheek-by-jowl with 2 novels and a film, and so many philosophical arguments, the Kondo discussion seems overlong. But there is also a risk in spending too much time elaborating the significance of this popular work.  Anyway I believe I have taken very seriously all the main points of the reviewer and am grateful for them.  I do hope the essay is better.